# Flow Modeling over Airfoils and Vertical Axis Wind Turbines Using Fourier Pseudo-Spectral Method and Coupled Immersed Boundary Method

Lucas Marques Monteiro *,† and Felipe Pamplona Mariano †

Laboratory of Thermal and Fluid Engineering (LATEF), School of Electrical, Mechanical and Computing Engineering (EMC), Federal University of Goiás (UFG), Av. Esperança, Al. Ingá, Prédio: B5, Campus Samambaia, Goiânia 74690-900, GO, Brazil
* Correspondence: lmmonteiro121@gmail.com
† These authors contributed equally to this work.

**Abstract:** In the present work, verifying the applicability and potentiality of the IMERSPEC methodology for numerical and computational modeling of two-dimensional flows over airfoils and vertical axis wind turbines is proposed. It is a high-order convergence methodology with low computational cost when compared to other high-order methods, resulting from the coupling of the Fourier pseudo-spectral method and the immersed boundary method. To validate the proposed methodology, flow simulations are carried out over an airfoil NACA 0012 for a Reynolds number equal to 1000. From the spatial discretization procedure, there is convergence and good agreement of the lift and drag coefficients and the Strouhal number in relation to reference works. The behavior of the flows over the airfoil, as a function of the angle of attack, is evaluated by pressure and vorticity fields. From the analyzed flows, it is observed that the formation of different wake modes, constituted by swirling structures that vary their characteristic sizes, is influenced by the angle of attack. A case study is proposed based on the analysis of the main fluid dynamic aspects of flows over wind turbines with a vertical axis of three blades for a Reynolds number equal to 100. For this, a mathematical model responsible for the imposition of the rotational movement on the blades is presented in the turbine. Performance parameters, such as the coefficient of tangential force and normal force, and the analysis of velocity fields on the simulated turbine were presented and compared with other numerical methods. The good level of convergence and the accuracy of the obtained results show the promising capacity of the IMERSPEC methodology in solving problems of this nature.

**Keywords:** flow over airfoil; vertical axis wind turbine; immersed boundary method; Fourier pseudo-spectral method

**MSC:** 76-10

## 1. Introduction

Airfoils are the two-dimensional sections of vertical axis turbine blades; they are also present in aircraft wings, engine blades and fan blades, with different geometric shapes and specific characteristics. Physically, the flow of a viscous fluid over an airfoil results in a fluid–structure interaction that promotes the appearance of a resultant force.

In aerodynamics, depending on the application, this force ensures the physical effect of lift and drag in large aircraft and in the well-known micro-aerial vehicles (MAVs) [1,2]. Among other physical mechanisms, it is also capable of promoting torque in wind turbines, responsible for maintaining the rotating movement of the blades and transforming the kinetic energy of the flow into mechanical energy [3]. In general, the design of these structures and the estimation of aerodynamic forces take into account the conditions imposed by the flow dynamics. In this sense, Computational Fluid Dynamics (CFD) has emerged as

a promising technique for research and development in various branches of engineering, especially in aeronautics, wind power, chemistry, petroleum and the environment.

Through numerical and computational methods, CFD proposes the solution of mathematical models formed by partial differential equations, such as the Navier–Stokes equations, which physically model the flows of a Newtonian fluid. In airfoils and vertical axis turbines, the solution of these models makes it possible to predict the performance and efficiency of these structures, verify the influence of design variables and allow the detailed visualization of velocity, pressure and vorticity fields [4–6]. They can be used as tools for systematic optimization procedures [7–10]. Furthermore, these are computational experiments, therefore, eliminating the need for calibrated and high-sensitivity measurement and control tools and instruments necessary in material experiments [11,12].

The application of CFD methodologies in the design, development and optimization of micro-air vehicles (MAVs) has shown potential results with physical consistency that are capable of being executed [13–15]. These flying objects, especially those with oscillating wings, are inspired by bees and insects and operate efficiently under very low Reynolds numbers ($Re \leq 10^3$). Low speeds and characteristic small sizes are factors that make them operational in difficult-to-access environments, allowing excellent maneuverability. In the case of fluid dynamics, the flows over the airfoils that make up the MAVs present important aspects to be investigated.

Using the finite volume method, [16] presented an analysis of the influence of the pitching motion on the formation of leading-edge vortices in an NACA 0012 airfoil subjected to flows with $Re = 3000$. A change in the evolution and behavior of the released structures was observed, implying sensitive variations in the aerodynamic forces applied to the airfoil. Two-dimensional flows over different symmetric NACA airfoils, with $Re$ = 400–6000, were modeled by [17,18] using the finite difference method and finite volume method, respectively. For the entire investigated range, there was evidence of early detachment of the flow and destruction of the laminar boundary layer, with the subsequent formation of a separation bubble on the upper surface of the airfoil, under the influence of different angles of attack. Using low-order convergence methods, [19] studied the dynamic stall phenomenon on a NACA 0012 airfoil for $Re = 1000$. Under high angles of attack, the formation of Kelvin–Helmholtz instabilities is observed, followed by a von-Kármán wake.

There is a trend toward the use of classical methodologies, such as the finite volume and finite difference method, for fluid dynamic computational modeling in airfoils and vertical axis turbines. The challenge for researchers is the search and development of models that are easily programmable, accessible, highly accurate, with low computational cost and with a high order of convergence. Boundary conditions in flows over airfoils and vertical axis wind turbines become complex as the industrial and engineering problems become increasingly sophisticated. Thus, paths are observed that allow the application of methods of a high order of convergence ($q > 2$), where $q$ is the order of convergence, in physical models of this nature, such as the Discontinuous Galerkin method, finite volumes of high order and spectral methods [20–22]. Unlike low-order convergence methods, it is possible to simulate complex physical problems, ensuring high accuracy without requiring very refined meshes, reducing cost and facilitating computational implementation.

Specifically, the Fourier pseudo-spectral method, presented in the present work, seems to be a method of a high order of convergence and excellent accuracy in relation to other methods, such as the method of low-order finite volumes, finite elements and finite differences. Mainly when applied to the solution of the set of Navier–Stokes equations and the continuity equation, transformed from physical space to spectral space [23–25]. High accuracy is guaranteed since spectral methods solve a derivative for a given point in the domain using information from all other points. The transformation operations are efficiently carried out using the Fast Fourier Transform (FFT) algorithm [26] and projection procedure, the pressure field is decoupled from the Navier–Stokes equations, being able to replace the solution of the linear system with a vector-matrix product to satisfy the conservation of mass.

The use of the FFT requires that the function to be transformed is necessarily periodic [27]. This condition is the biggest limitation of the Fourier pseudo-spectral method, especially for fluid dynamics problems. Flows over airfoils and mobile geometries, such as the vertical turbine blades under rotating motion or flows over deformable geometries, require the imposition of any boundary conditions different from periodic conditions. Therefore, for the modeling of flows with non-periodic conditions using the pseudo-spectral method, it is necessary to apply additional mathematical and numerical tools. Thus, the application of the immersed boundary method [28,29] is proven to be an efficient alternative with low computational cost to overcome this limitation compared with other methods that make use of unstructured meshes.

The ability of the boundary-immersed methods in complex models and mobile geometries allows its application in problems involving flows over airfoils and vertical axis wind turbines. A fluid–structure problem was proposed by [30] based on a flow over an oscillating airfoil. It was observed in the results that despite the complexity of the problem, the modeling presented an economy in its computational cost by the use of the immersed boundary method. Two-dimensional and three-dimensional flow simulations over an NACA 0012 airfoil were performed by [31] using different types of immersed boundary methods associated with the finite volume method. The presented results validated the applicability of the immersed boundary method to represent, with reliability and good levels of detail, the complex movement of a flapping wing flight. Refs. [32–34] studied the application of the immersed boundary method, associated with the large-eddy simulation model (LES), in simulations of flows over airfoils and vertical axis turbines. The complexity and non-linearity of the flows, due to the turbulent regime, are well represented, especially in the wake region, advected and in the regions adjacent to the boundary layer.

The hybridization of the Fourier pseudo-spectral method and the immersed boundary method resulted in the IMERSPEC methodology [25,35,36]. This coupling added the advantages of both methods, with emphasis on the modeling of flows over complex and mobile geometries using a Cartesian mesh under non-periodic boundary conditions; the decoupling of pressure-velocity variables, eliminating the need to solve the Poisson equation; post-processing for the recovery of the pressure field, satisfying the continuity equation with round-off errors.

Based on this approach, the present work proposes the evaluation of the applicability and potentiality of the IMERSPEC methodology for the solution of two-dimensional and incompressible flows over airfoils and blades of vertical axis turbines in rotating motion under low Reynolds numbers ($Re \leq 10^3$). For this, it presents the development of a specific subroutine capable of modeling the rotating movement of the blades. Associated with the IMERSPEC methodology, the complete model is capable of contemplating the fluid–structure interaction between a turbine and flow. It also proposes numerical procedures that allow the calculation of the main performance parameters of airfoils and vertical axis turbines, among which the following stand out: coefficient of lift, drag, normal and tangential.

## 2. Mathematical Modeling

This section aims to detail the IMERSPEC methodology [25], used in this work as a numerical and computational tool for modeling and simulating flows over airfoils and blades of vertical axis turbines. Initially, the mathematical modeling of fluid flows in physical space is addressed. Then, using Fourier's pseudo-spectral method, the transformation of the equations from the physical space to the spectral space is presented, and the formulation of the immersed boundary method, based on the interactive process of Multi-Direct Forcing (MDF). Finally, the development of the mathematical model that imposes movement to the blades of a vertical axis turbine is shown, simulating the rotating effect of the rotor.

### 2.1. Mathematical Modeling of Fluid Flows

The fluid dynamic behavior of flows over airfoils and vertical axis turbine blades is described by a differential mathematical model composed of the continuity equation, given by Equation (1), and by the Navier–Stokes equations, given by Equation (2). This model is presented in tensorial notation, valid for $t \geq 0$,

$$\frac{\partial u_j}{\partial x_j} = 0, \tag{1}$$

$$\frac{\partial u_i}{\partial t} + \frac{\partial (u_i u_j)}{\partial x_j} = -\frac{\partial p}{\partial x_i} + \nu \frac{\partial^2 u_i}{\partial x_j \partial x_j} + f_i, \tag{2}$$

where $t$ is the time, $u_i(\mathbf{x}, t)$ are the components of the velocity vector in (m/s), $\mathbf{x}$ is the position vector of a point in the Eulerian domain, $p = p^*/\rho$, where $p^*$ is the static pressure field in (N/m²), $\rho$ is the specific mass of the fluid in (kg/m³) and $\nu$ is the kinematic viscosity of the fluid in (m²/s). The term $f_i = f_i^*/\rho$ is used to model the components of any force field applied to the flow, where $f_i^*$ is shown in (N/m³). These force fields are due to fluid–structure and fluid–fluid interactions, due to electromagnetic effects, gravitational effects or other physical effects internal to fluid particles.

Using the immersed boundary method, the term $f_i$ virtually models the immersed interface of airfoils and vertical turbine blades. In the present work, the mathematical model is restricted to two-dimensional, incompressible, isothermal flows, Newtonian fluids and constant fluid physical properties.

### 2.2. Fourier Pseudo-Spectral Method

The Fourier pseudo-spectral method is responsible for transforming the primitive variables of fluid dynamics (velocity and pressure) from physical space to spectral space using the direct Fourier transform, Equation (3), and the inverse Fourier transform, Equation (4),

$$\widehat{\sigma}(\mathbf{k}, t) = \left(\frac{1}{2\pi}\right)^2 \int_{-\infty}^{\infty} \sigma(\mathbf{x}, t) e^{-\iota \mathbf{k}\mathbf{x}} d\mathbf{x}, \tag{3}$$

$$\sigma(\mathbf{x}, t) = \left(\frac{1}{2\pi}\right)^2 \int_{-\infty}^{\infty} \widehat{\sigma}(\mathbf{k}, t) e^{\iota \mathbf{k}\mathbf{x}} d\mathbf{k}, \tag{4}$$

where $\widehat{\sigma}(\mathbf{k}, t)$ is the field of the transformed variable, $\sigma(\mathbf{x}, t)$ is the untransformed variable field, $\mathbf{k}$ is the wavenumber vector and $\iota = \sqrt{-1}$ is the imaginary number. Equation (3) obtains the field of the variable in the spectral space $\widehat{\sigma}(\mathbf{k}, t)$, transformed from physical space. By Equation (4), we obtain the field of the variable in physical space $\sigma(\mathbf{x}, t)$, transformed from the spectral space.

Fourier's pseudo-spectral method, therefore, allows the transformation of the differential mathematical model, described in Section 2.1, from physical space to spectral space (Fourier space). Thus, applying Equation (3) to the continuity equation, Equation (1), we have,

$$\iota k_j \widehat{u}_j = 0, \tag{5}$$

where $\widehat{u}(\mathbf{k}, t)$ is the velocity field in spectral space.

The transformation of Equation (2) to Fourier space is given by,

$$\frac{\partial \widehat{u}_i}{\partial t} + \iota k_j \widehat{(u_i u_j)} = -\iota k_i \widehat{p} - \nu k^2 \widehat{u}_i + \widehat{f}_i, \tag{6}$$

where $k^2 = k_j k_j$ is the squared norm of the vector wavenumber $\mathbf{k}$.

In spectral space, the non-linear term $\iota k_j \widehat{(u_i u_j)}$ is given by the product of two transformed functions $\widehat{(u_i u_j)}$. Formally, solving this term requires solving a convolution integral, such that the transformed non-linear term becomes,

$$\iota k_j \widehat{\widehat{u_i u_j}}(\mathbf{k}) = \iota k_j \int\limits_{\mathbf{k}=\mathbf{r}+\mathbf{s}} \hat{u}_i(\mathbf{r})\hat{u}_j(\mathbf{k}-\mathbf{r})d\mathbf{r}, \tag{7}$$

where $\mathbf{k} = \mathbf{r} + \mathbf{s}$, gives the triadic interactions between the vector's wavenumber $\mathbf{k}$, $\mathbf{r}$ e $\mathbf{s}$.

One can, therefore, rewrite Equation (6),

$$\left[\frac{\partial}{\partial t} + \nu k^2\right]\hat{u}_i(\mathbf{k}, t) = \wp_{im}\left[\widehat{f}_m(\mathbf{k}, t) - \iota k_j \int\limits_{\mathbf{k}=\mathbf{r}+\mathbf{s}} \widehat{u}_m(\mathbf{r}, t)\hat{u}_j(\mathbf{k}-\mathbf{r}, t)d\mathbf{r}\right], \tag{8}$$

where $\wp_{im}$ is the projection tensor, defined by,

$$\wp_{ij}(\mathbf{k}) = \delta_{ij} - \frac{k_i k_j}{k^2}, \tag{9}$$

where

$$\delta_{ij} = \left\{\begin{array}{ll} 1 & se \quad i = j \\ 0 & se \quad i \neq j \end{array}\right., \tag{10}$$

is the Kronecker delta. The tensor $\wp_{ij}$ projects any vector on the plane $\pi$. This is the plane of zero divergence defined in the spectral space, perpendicular to the vector $\mathbf{k}$. For more details, see the works [25,37].

Note that the pressure term in spectral space $-\iota k_i \hat{p}$ becomes null, projected by the tensor $\wp_{im}$. This way, the pressure-velocity coupling is eliminated and, consequently, the need to solve the Poisson equation, commonly proposed in classical numerical methods, such as the finite volume method [38]. The velocity field is now determined without necessarily solving the pressure field, and mass conservation is guaranteed by a vector-matrix product, which is computationally cheaper compared to the interactive procedure of the Poisson equation, which requires the solution of a linear system.

The recovery of the pressure field is given through post-processing, from which it is calculated by,

$$\hat{p}(\mathbf{k}) = \frac{\iota k_m}{k^2}\left[-\widehat{f}_m(\mathbf{k}) + \iota k_j \int\limits_{\mathbf{k}=\mathbf{r}+\mathbf{s}} \widehat{u}_m(\mathbf{r})\hat{u}_j(\mathbf{k}-\mathbf{r})d\mathbf{r}\right]. \tag{11}$$

In addition to the projection procedure, another computational procedure used in the methodology is the elimination of the need to solve the convolution integral of the non-linear term in the spectral space, shown in Equation (7). The solution of this integral requires the application and implementation of numerical integration schemes, making the proposed methodology onerous. To overcome this drawback, ensuring the accuracy of the methodology, the calculation of the integral is replaced by the product of two functions in physical space and only then is the direct Fourier transform applied to the product. This way, the non-linear term of the Navier–Stokes equations is solved pseudo-spectrally.

Therefore, it is avoided to calculate the product, in the spectral space, of two individually transformed functions, which mathematically requires a solution of the convolution integral. For example, given a function $g(\mathbf{x}, t)$ and $h(\mathbf{x}, t)$, in the physical space, the product is made in such a way that: $b(\mathbf{x}, t) = g(\mathbf{x}, t)h(\mathbf{x}, t)$. We apply Equation (3) in $b(\mathbf{x}, t)$, obtaining $\hat{b}(\mathbf{k}, t)$. The same treatment is applied to the non-linear term.

With this procedure, the accuracy of the spectral method is maintained since the derivatives are still calculated in the spectral space, and the low computational cost acquired with the pressure field projection procedure is maintained. However, the number of times the direct and inverse Fourier transform is applied to the variables is increased at each time step, especially when treating the non-linear term in the antisymmetric form [23], as performed in the present work. The non-linear term in its antisymmetric form is given

from an arithmetic mean of the conservative and non-conservative form and guarantees greater numerical stability.

The Fourier pseudo-spectral method is based on the application of Equations (3) and (4), in their discrete versions, by using the Fast Fourier Transform (FFT) [26]. This requires that the functions to be transformed are necessarily periodic. The immersed boundary methodology proved to be an efficient computational alternative to circumvent this limitation, from which the flow can be well resolved and the continuity equation satisfied in complex regions of the geometry without becoming onerous and computationally expensive.

### 2.3. Immersed Boundary Method

The immersed boundary method uses two simultaneous and independent calculation domains, the Eulerian domain ($\Omega$), fixed and Cartesian, and the domain that delimits the interface immersed in the flow, called the Lagrangian domain ($\Gamma$), as shown in Figure 1.

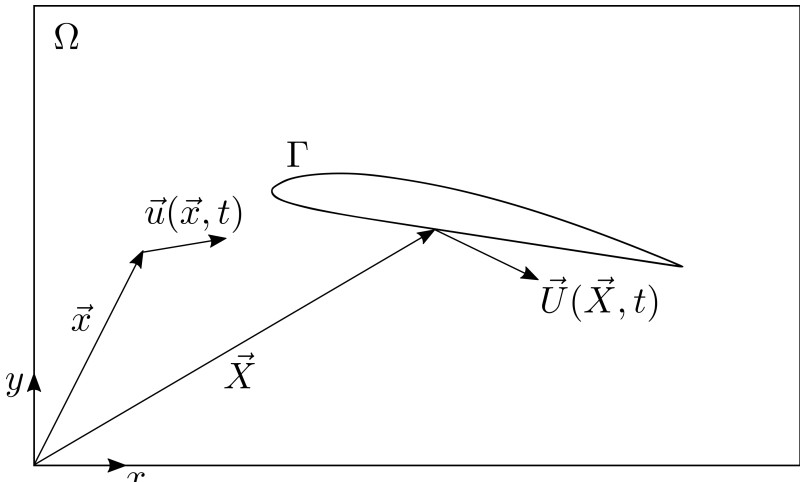

**Figure 1.** Representation of the Eulerian domain ($\Omega$) and the Lagrangian domain ($\Gamma$), where **x** is the position vector of any point in the Eulerian domain and **X** is the position vector of any point in the Lagrangian domain.

The mathematical model presented in Section 2.1 is solved and valid for the entire Eulerian domain, including the region delimited by the Lagrangian domain. In the present work, as it is a two-dimensional analysis, the Lagrangian domain is represented by the contour lines that model the geometry of the airfoil and the blades of a vertical turbine, constituted by discrete points.

The coupling procedure between the Eulerian and Lagrangian domains is given mathematically by calculating the force term $f_i$, presented in Equation (2). From this calculation, using previously determined Lagrangian points as a reference, the interface immersed in the Eulerian domain is virtually modeled. It is, therefore, the imposition of boundary conditions on a frontier, informing the flow of the presence of a body immersed in it.

The contribution of the term $f_i$ is important near the interface, determined by a discontinuous function. Thus, over every Eulerian domain, $f_i$ is zero except when approaching the Lagrangian points,

$$f_i(\mathbf{x}, t) = \sum_{\Gamma} D_h(\mathbf{x} - \mathbf{X}) F_i(\mathbf{X}, t) \Delta s^2, \tag{12}$$

where $F_i(\mathbf{X}, t)$ is the Lagrangian force, $\Delta s$ is the spacing between the discretized Lagrangian points and $D_h(\mathbf{x} - \mathbf{X})$ is a distribution function,

$$D_h(\mathbf{x} - \mathbf{X}) = \frac{1}{\Delta x^2} W_h(r_x) W_h(r_y), \tag{13}$$

where $r_x = \frac{x-X}{\Delta x}$, $r_y = \frac{y-Y}{\Delta y}$, $\Delta x$ and $\Delta y$ are the spacing between the discretized Eulerian points in the $x$ and $y$ directions, respectively, and $W_h$ is the "hat" weight function, calculated by,

$$W_h(r) = \begin{cases} 1 - |r|, & \text{se } 0 \le |r| \le 1 \\ 0, & \text{se } 1 < |r| \end{cases}. \tag{14}$$

The complexity of the geometric shapes of the airfoils makes it difficult to coincide between the Lagrangian points and the Eulerian points. Therefore, the Lagrangian force distribution procedure becomes necessary. We calculate the Lagrangian force field $F_i(\mathbf{X}, t)$, which is distributed to the nearest neighboring Eulerian points using Equation (12), together with Equations (13) and (14).

The choice of the appropriate weight function guarantees the high accuracy and order of convergence of the numerical solutions. Other functions can be found in [28,39]. In simulations of flows over airfoils, the results obtained by the "hat" function were more accurate, to the detriment of the other functions. These results are explained due to the use of only three points in the support function. While other functions require at least five points. As the geometry of an airfoil is slender, especially at the trailing edge, there is no space for weight functions that require many points for the respective velocity interpolations and force distributions.

The immersed boundary methodology, based on multiple direct force imposition, requires the calculation of the Lagrangian force $F_i(\mathbf{X}, t)$.

Starting from Equation (2), one isolates $f_i$,

$$f_i(\mathbf{x}, t) = \frac{u_i^{t+\Delta t} - u_i^* + u_i^* - u_i^t}{\Delta t} + rhs_i^t, \tag{15}$$

where $u_i^{t+\Delta t}$ is the velocity component of an Eulerian point at the current instant of time $t + \Delta t$, $u_i^t$ is the velocity component of an Eulerian point at the previous instant of time $t$, $u_i^*$ is the estimated Eulerian point velocity component, $\Delta t$ is the discretized time step and $rhs_i$ is the sum of the diffusive, advective (non-linear term) and pressure gradient terms, which set Equation (2).

Two details presented by Equation (15) are worth highlighting. First, the time derivative was discretized by the explicit Euler method. This choice was merely explanatory and didactic for understanding the modeling. The temporal discretization in the present work was performed using the fourth-order Runge–Kutta method of six-step temporal convergence (RK46) [40]. According to the Eulerian velocity $u_i^*$ is a temporary parameter, understood as the estimate of the Eulerian velocity field at the current time step without taking into account the correction by the force source term $f_i$.

Decomposing Equation (15) into two parts,

$$\frac{u_i^* - u_i^t}{\Delta t} + rhs_i^t = 0, \tag{16}$$

$$f_i(\mathbf{x}, t) = \frac{u_i^{t+\Delta t} - u_i^*}{\Delta t}. \tag{17}$$

By the continuum hypothesis, Equation (17) can be defined either in the Eulerian domain ($\Omega$) or in the Lagrangian domain ($\Gamma$). Therefore, the Lagrangian force $F_i(\mathbf{X}, t)$,

$$F_i(\mathbf{X}, t) = \frac{U_i^{t+\Delta t} - U_i^*}{\Delta t}, \tag{18}$$

where $U_i^{t+\Delta t}$ is the velocity component of the Lagrangian points that models the immersed boundary in the time step $t + \Delta t$, therefore, $U_i^{t+\Delta t} = U_i^{FI}$. For the case of flows over airfoils, $U_i^{FI} = 0$ throughout the simulated physical time. For flows over the blades of vertical axis turbines, $U_i^{FI}$ is given by a mathematical model that imposes rotary movement on the interface of the blades.

Initially, to determine the temporary parameter $U_i{}^*$, Equation (16) is solved. From this solution, we obtain $u_i{}^*$. Then, the opposite process to the distribution of the Lagrangian force, called velocity interpolation, is carried out $u_i{}^*$,

$$U_i{}^*(\mathbf{X}, t) = \sum_{\Omega} D_h(\mathbf{x} - \mathbf{X}) u_i{}^*(\mathbf{x}, t) \Delta x^2. \tag{19}$$

By Equation (19), the information of interest of the Eulerian domain $u_i{}^*$ is transferred to the Lagrangian domain. Just like the distribution procedure, in interpolation, the Eulerian information is passed on to the closest neighboring Lagrangian points, weighted by the distance between these points. Thus, the Lagrangian points closer to the Eulerian points present greater contributions to Eulerian information, while the more distant ones present smaller contributions.

The Lagrangian force field, calculated by Equation (18), is distributed, as shown in Equations (12)–(14). Rearranging Equation (17), term $f_i$, obtained from the distribution procedure, is corrected to $u_i{}^*$ and is updated to $u_i{}^{t+\Delta t}$,

$$u_i{}^{t+\Delta t} = u_i{}^* + \Delta t f_i. \tag{20}$$

Thus far, the details described by the method are responsible for ensuring the imposition of the boundary condition on the immersed interface by the force term, $f_i$. Therefore, in order to improve the understanding, the steps taken to impose this condition are organized in sequence:

1.  Using Equation (16), we calculate the parameter temporary or Eulerian velocity field $u_i{}^*$;
2.  By the interpolation procedure presented by Equation (19), the information from the Eulerian domain, $u_i{}^*$, is transmitted to the Lagrangian domain. Thus, $U_i{}^*$ is determined;
3.  $U_i{}^{FI}$ is determined, that is, the velocity that the immersed boundary must have over the simulated physical time. This velocity is imposed or calculated by some additional mathematical model. For airfoils, $U_i{}^{FI} = 0$. For vertical axis turbine blades, $U_i{}^{FI}$ is calculated by Equations (25) and (26), described in Section 2.4;
4.  Using Equation (18), we calculate the Lagrangian force. In general, this step is about the application of Newton's Second Law on the Lagrangian domain. The boundary condition of the immersed interface, given by $U_{FI}$, is now guaranteed in terms of the Lagrangian force;
5.  Using Equation (12), we propose the distribution of the Lagrangian force for the points of the Eulerian domain. The Eulerian force field $f_i$ is determined;
6.  The Eulerian velocity field $u_i{}^*$ is then corrected by the term $f_i$, using Equation (20).

In summary, at the end of step (6), the flow is informed, indirectly, of the velocity boundary condition, imposed on the boundary in step (3), by the Eulerian force field $f_i$, which corrects the velocity field $u_i$.

Numerically, the temporal discretization procedures of the IMERSPEC methodology and the use of the distribution and interpolation functions for the virtual modeling of the immersed body are responsible for absolutely not satisfying the non-slip condition. To get around this limitation, an interactive procedure is used to improve the accuracy of the velocity calculation $u_i{}^{t+\Delta t}$. This iterative procedure is the Multi-Direct Forcing method [25,29], described by the steps below:

1.  Before advancing the time step, $u_i{}^{t+\Delta t, it} = u_i{}^{t+\Delta t}$, where $it$ is the interaction of multiple direct imposition of force;
2.  It is interpolated to $u_i{}^{t+\Delta t, it}$, using Equation (19);
3.  Obtained from interpolation, the new Lagrangian velocity $U_i{}^{t+\Delta t, it}$ is replaced in Equation (18). It is calculated as $F_i^{it}$;
4.  Using Equations (12)–(14), we distribute $F_i^{it}$, and it is obtained as $f_i^{it}$;

5.　With the term $f_i^{it}$, it is corrected to $u_i^{t+\Delta t,it}$, and it is estimated as $u_i^{t+\Delta t,it+1}$,

$$u_i^{t+\Delta t,it+1} = u_i^{t+\Delta t,it} + \Delta t f_i^{it}; \tag{21}$$

6.　It is updated to $it = it + 1$, return to step (1) or advance in time, $t + \Delta t$.

In the present work, the described interactive process is carried out until a maximum number of interactions (*NIT*), configured in the initial setup of the IMERSPEC methodology algorithm, are satisfied. Therefore, when reaching *NIT* interactions, the interactive loop is completed, and the model advances in time.

The greater the number of interactions performed, the more accurate the imposition of the non-slip condition on the surface of the immersed bodies becomes. It results, therefore, in greater accuracy in determining the performance parameters of airfoils and vertical axis turbine blades, such as: coefficients of lift, drag, power, normal, tangential, among others. However, the computational cost of the methodology becomes proportional to the increase in the number of interactions.

The hybridization of the Fourier pseudo-spectral method and the immersed boundary method results in the IMERSPEC methodology. Details of the coupling procedure between the two methodologies are described in [25].

### 2.4. Mathematical Modeling of Rotary Motion

In the immersed boundary method, the Eulerian domain is Cartesian and remains fixed throughout the simulated physical time. For fluid dynamic problems involving rotating boundaries, it is necessary to impose a mathematical model on the Lagrangian domain that allows calculating, at each time step $\Delta t$, a new position to the Lagrangian points from their initial positions. In the present work, this model simulates the rotational movement of vertical wind turbine blades.

The Lagrangian points, which together delimit the interface of the airfoils, assume new positions at each instant of time due to the speed of rotation $\omega$ of the turbine.

In Figure 2, the scheme of the rotational movement of a Lagrangian point is shown, belonging to the domain $\Gamma$. The Lagrangian point, represented by $p^{t_0}$, at the initial time $t_0$, is shifted $\phi^t$, where $\phi^t$ is the angle of rotation promoted by the imposition of $\omega$. Thus, this point assumes a new position at an instant of time $t$, represented by $p^t$. The location of the Lagrangian point in relation to the center of rotation of the turbine, at the instants of time $t_0$ and $t$, is given by the position vector $\mathbf{X}^{t_0}$ and $\mathbf{X}^t$, respectively. The axis system $xy$ established the origin of the Eulerian domain and is positioned at the center of the rotor.

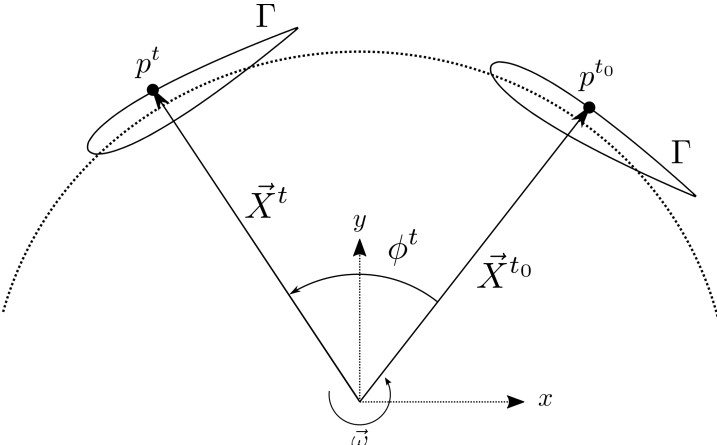

**Figure 2.** Rotary motion of a Lagrangian point $p$, imposed by a speed of rotation $\omega$.

To determine the positions of the Lagrangian points, imposed by the rotary motion, is to locate them by the position vector $\mathbf{X}^t$ at every instant of time $t$. Therefore, the position

$\mathbf{X}^t$ of a given point, starting from the origin of the Eulerian domain, is calculated, starting from its initial position $\mathbf{X}^{t_0}$,

$$\mathbf{X}^t = S^\phi \mathbf{X}^{t_0}, \tag{22}$$

where $S^\phi$ is the rotation matrix,

$$S^\phi = \begin{pmatrix} cos(\phi^t) & -sin(\phi^t) & 0 \\ sin(\phi^t) & cos(\phi^t) & 0 \\ 0 & 0 & 1 \end{pmatrix}. \tag{23}$$

The matrix, given by Equation (23), accounts for the effect of turbine rotation on a given Lagrangian point, changing the direction of the initial position vector $\mathbf{X}^{t_0}$ from the angle of rotation $\phi^t$,

$$\phi^t = \omega(t - t_0), \tag{24}$$

where $\omega$ is the magnitude of the rotation speed of the turbine, positive in the counterclockwise direction.

Given a Lagrangian point, subjected to $\omega$, the new position $\mathbf{X}^t$ assumed by that point is displaced $\phi^t$ relative to the starting position $\mathbf{X}^{t_0}$, and is calculated by Equation (22). This procedure is performed at each instant of time $t$ for all the Lagrangian points that make up the airfoils of the blades, simultaneously displacing them in relation to their initial positions. Physically, this models the rotating motion of the turbine rotor blades.

In addition to speed $\omega$, another input data for the calculation of the new positions is the pre-determination of the initial position of all Lagrangian points $\mathbf{X}^{t_0}$. In the computational platform, the insertion of these data is carried out in the configuration phase of the simulation parameters.

The calculation of the Lagrangian force, Equation (18), requires the determination of the tangential velocity components $U^{FI}$ of each Lagrangian point that represents the immersed boundary, shown in Figure 3b and calculated by

$$U_x^{FI} = -\omega X sin(\varphi^{FI}), \tag{25}$$

$$U_y^{FI} = \omega X cos(\varphi^{FI}), \tag{26}$$

where $X = \sqrt{X_p{}^2 + Y_p{}^2}$ is the magnitude of the position vector $\mathbf{X}$, in which $X_p$ is the coordinate of $\mathbf{X}$ in the $x$-direction and $Y_p$ is the coordinate of $\mathbf{X}$ in the $y$-direction. The angle $\varphi^{FI}$ is positive counterclockwise,

$$\varphi^{FI}(\theta) = \begin{cases} 180° + tan^{-1}\left(\dfrac{Y_p}{X_p}\right), & se\ 0° \leq \theta \leq 180° \\ tan^{-1}\left(\dfrac{Y_p}{X_p}\right), & se\ 180° < \theta \leq 360° \end{cases}, \tag{27}$$

and estimated as a function of the azimuthal position $\theta$, where the Lagrangian point lies, illustrated in Figure 3a.

In the present work, the origin of the Eulerian domain is positioned at the center of the rotor. Then, note that determining $U^{FI}$ is similar to the procedure for estimating the tangential blade velocity as a function of the turbine radius $R$. However, in the presented modeling, the calculation of the tangential velocity is given point by point, in a localized way, at the level of Lagrangian points, weighted by the position $\mathbf{X}$ of each one of them.

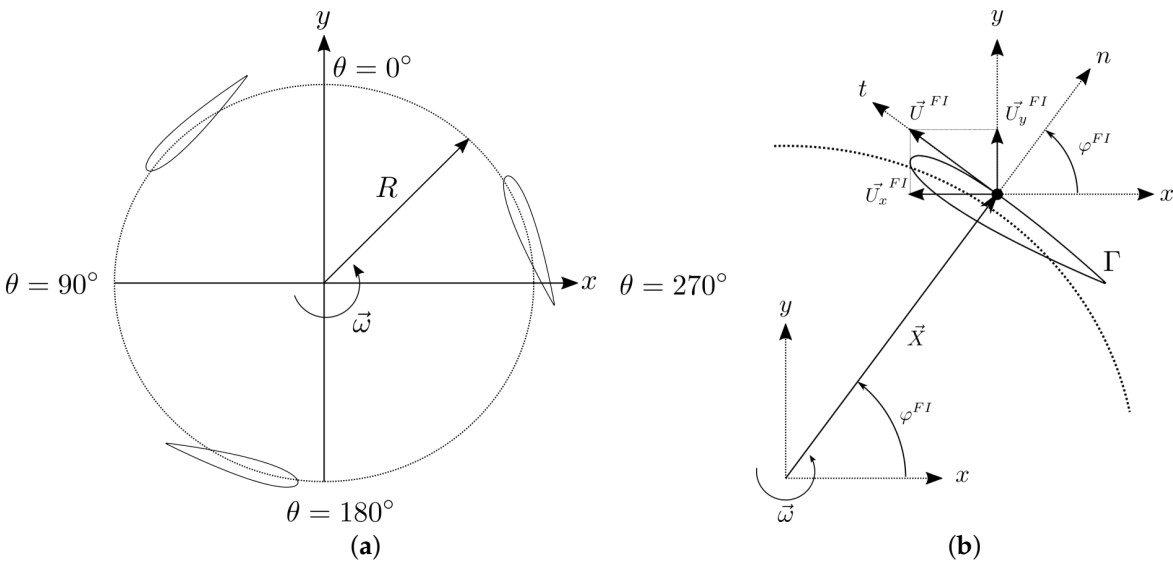

**Figure 3.** (**a**) Top view of the rotor of a three-blade vertical axis turbine: azimuthal angle variation $\theta$. (**b**) Imposition of the tangential velocity on a Lagrangian point belonging to the blade of a vertical axis turbine.

As this is a two-dimensional analysis, Figure 4 shows the two vector components of the Lagrangian force, $F_x$ and $F_y$, that act in the $x$ and $y$ directions, respectively. These forces are calculated by Equation (18) from the tangential velocity components over each Lagrangian point, determined by Equations (25) and (26). The decomposition of forces $F_x$ and $F_y$, from angle $\varphi^{FI}$, results in the calculation of the tangential force $F_t$ acting in direction $t$ and at normal force $F_n$ acting in the direction $n$. In magnitude, it is calculated by,

$$F_t = -F_x sin(\varphi^{FI}) + F_y cos(\varphi^{FI}), \tag{28}$$

$$F_n = F_x cos(\varphi^{FI}) + F_y sin(\varphi^{FI}), \tag{29}$$

respectively.

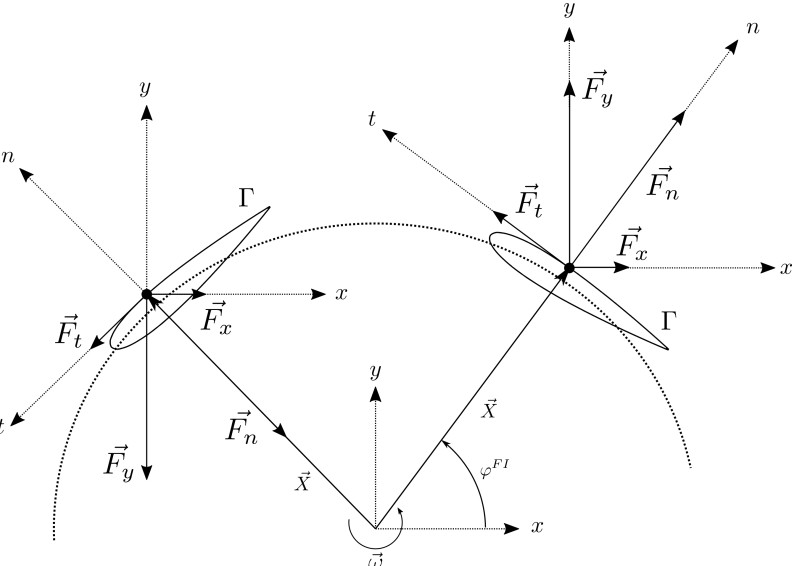

**Figure 4.** Decomposition of the components of the Lagrangian force: tangential force $F_t$ and normal force $F_n$ applied to a Lagrangian point.

Using Equations (28) and (29), it is proposed to calculate the components of tangential and normal forces individually on each Lagrangian point. The global contributions, that is, the net effect of the force components applied to the immersed body, are represented in Figure 5. In vertical axis turbines, we know the tangential force applied to the blade $F_T$ is fundamental to evaluating the necessary torque responsible for the transformation of the kinetic energy of the fluid into mechanical energy. Through normal force $F_N$, the structural loads under which the turbine must withstand due to the interaction of the blades with the flow are evaluated, as shown in Figure 5a. In flows over airfoils, Figure 5b, without imposing rotational motion at the boundary, two global performance forces are important: the lift force $F_L$ and the drag force $F_D$.

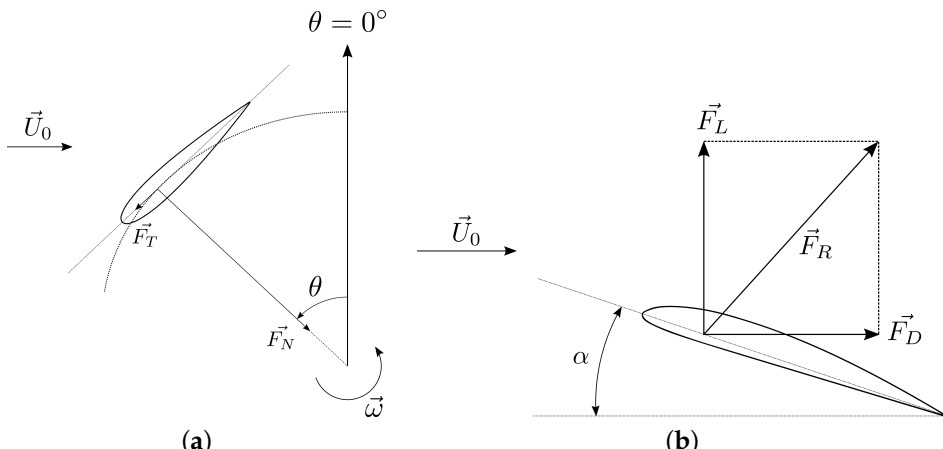

**Figure 5.** (**a**) Components of the forces applied to the blade of a vertical axis turbine, subjected to a flow with free stream velocity $U_0$. (**b**) Components of the forces applied to an airfoil, where $\alpha$ is the angle of attack.

Mathematically, in the immersed boundary method, the global contributions $F_T$, $F_N$, $F_L$ and $F_D$ are obtained by,

$$F_{ci} = -\rho \sum_{it=1}^{NIT} \sum_{p=1}^{N_L} F_i^p(\mathbf{X}, t)\Delta x \Delta s, \tag{30}$$

where $F_{ci}$ is calculated, in $[N]$, by the sum of a given force component $F_i^p$ ($F_n$, $F_t$, $F_x$ or $F_y$) applied over all Lagrangian points, along all Multi-Direct Forcing interactions and $N_L$ is the number of Lagrangian points that model the discretized embedded interface. Note that the length of the third direction is unity.

The magnitudes of these force components are quantified by their respective dimensionless coefficients, calculated by,

$$C_i = \frac{2F_{ci}}{\rho U_0^2 c(1)} \tag{31}$$

where $U_0$ is the free stream velocity of the flow in (m/s) and $c$ is the chord of the airfoil or blade of a vertical axis turbine in (m).

Therefore, to determine $F_T$ and $F_N$ on the blade of a turbine at a given instant in time $t$, the calculation is carried out using $F_t$ and $F_n$ for each Lagrangian point and along each Multi-Direct Forcing interaction, using Equations (28) and (29), respectively. These parcels are then added to Equation (30). To determine $F_D$ and $F_L$, applied to an airfoil, the same procedure is performed, in which the sum is given by the individual contributions of the Lagrangian forces $F_x$ and $F_y$ on each point, respectively, calculated by Equation (18).

### 3. Numerical and Computational Modeling

The transformation of the primitive variables of the mathematical model (pressure and velocity) from the physical domain to the spectral domain is numerically performed by the Discrete Fourier Transform (DFT) version using the Fast Fourier Transform (FFT) algorithm [26].

The FFT efficiently operates by the bit rotation procedure, which decreases the number of bit operations $\mathcal{O}(N^2)$ to $\mathcal{O}(Nlog_2N)$, where $N$ is the number of collocation points of the discrete Eulerian domain. In the present work, we use the two-dimensional version of FFTE, which is available in [41]. The algorithm was written in FORTRAN 77 programming language.

In the physical domain, the variables of the mathematical model are spatial. In the spectral domain, the transformed variables are defined as a function of **k**, wavenumber vectors, defined by,

$$k_i(n) = \begin{cases} \frac{2\pi}{L_i}(n-1) & 1 \leqslant n \leqslant \frac{N_i}{2}+1 \\ \frac{2\pi}{L_i}(n-1-N_i) & \frac{N_i}{2}+2 \leqslant n \leqslant N_i \end{cases}, \tag{32}$$

where $k_i$ is component $i$ of the wave number vector, $N$ is the number of placement nodes in a given direction, $L$ is the physical length of the domain in that given direction and $n$ is the position in the vector in a given direction of the domain.

Temporal discretization is performed by the fourth-order Runge–Kutta method of temporal convergence, with six steps (RK46), presented by [40]. It is a high-order convergence method, optimized, with reduced variable storage cost, low dispersion and low numerical dissipation. The time increment $\Delta t$ is calculated in a variable way,

$$\Delta t = CFL.min\left[min\left[\frac{\Delta x}{max[|u|]}; \frac{\Delta y}{max[|v|]}\right]; \frac{2}{\nu}\left(\frac{1}{\Delta x^2} + \frac{1}{\Delta y^2}\right)^{-1}\right], \tag{33}$$

where $CFL$ is a parameter between 0 and 1, configured in the initial setup of the program according to the type of flow being analyzed.

The CFL parameters established in the present work, $CFL = 0.25$ to flows over airfoil and $CFL = 0.10$ to flows over vertical axis turbine, are well below the limits of accuracy and stability proposed by the temporal discretization procedure, as shown by [40]. This means minimizing the temporal errors present in the results as much as possible. Therefore, these results are allowed to present errors, mainly arising from the spatial discretization procedure.

### 4. Validation

In this section, the present work proposes to evaluate the ability of the IMERSPEC methodology to model a fluid dynamic problem based on incompressible flows over an airfoil.

### 4.1. Calculation Domain

Figure 6 shows the domain used for simulations of flows over a NACA 0012 airfoil. This domain is divided into two main regions: the calculation domain ($\Omega_P$) and the physical domain ($\Omega_{nP}$). The boundary $\Gamma_P$ delimits $\Omega_P$, where the periodic boundary conditions required by the Fourier pseudo-spectral method are imposed. The physical domain, delimited by the boundary $\Gamma_{nP}$ is, in fact, where the fluid dynamics of the flow are simulated, and the results of the physical problem are obtained. In the boundary $\Gamma_1$, formed by the Lagrangian points that represent the airfoil, the no-slip boundary condition is imposed by the term $f_i(\vec{x}, t)$, with $U_i^{FI} = 0,00$ (m/s).

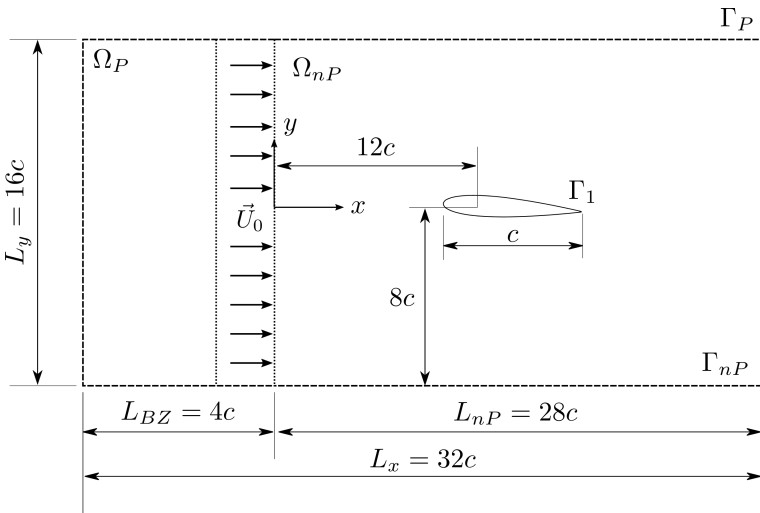

**Figure 6.** Calculation domain for the solution of flows over an airfoil.

Domain dimensions are dimensionalized by the airfoil chord, $c = 1.00$ (m). The imposition of the periodic boundary condition on the output region of the domain leads to the reinjection of physical instabilities in the input region of the calculation domain. These instabilities are smoothed out by the buffer zone [25]. In addition, the domain contains the forcing zone that guarantees the alignment of the flow in the entrance region of the physical domain and directs the entrance velocity profile horizontally [25]. The imposition of the velocity in the forcing zone is performed by the term $f_i(\vec{x}, t)$. Together, the buffer zone and the forcing zone have a length of $L_{BZ} = 4c$. The physical domain has a length of $L_{nP} = 28c$. The total length of the calculation domain is $L_x = 32c$.

The simulations are performed for a NACA 0012 airfoil with a rounded trailing edge. The Reynolds number of the flow is equal to $Re = 1000$. The kinematic viscosity of the fluid is calculated by $\nu = U_0 c / Re$, in (m$^2$/s). The specific mass of the fluid is equal to $\rho = 1.00$ (kg/m$^3$). The airfoil is subjected to a free stream velocity $U_0 = 1.00$ (m/s).

The time increment $\Delta t$ is defined by $CFL = 0.25$; see Equation (33). The total range of simulated physical time is $t^* = [0 : 100]$, where $t^* = tU_0/c$ is dimensionless time.

### 4.2. Eulerian Domain Refinement

The Eulerian domain mesh refinement test is performed for three meshes: $N_x = 512 \times N_y = 256$, $N_x = 1024 \times N_y = 512$ and $N_x = 2048 \times N_y = 1024$, where $N_x$ is the number of colocation points that discretizes the length $L_x$ and $N_y$ is the number of colocation points that discretizes the length $L_y$. The airfoil is discretized into $N_L = 300$ Lagrangian points, spaced by the $\Delta s$ variable. The maximum number of interactions is $NIT = 50$. The airfoil is positioned at angles of attack $\alpha = 10°$ and $16°$. The influence of mesh refinement on the lift and drag coefficients is analyzed and presented in Figure 7.

Mesh refinement imposes a significant change in the transient behavior of $C_l$ and $C_d$. In Figure 7a,b, it is observed that finer meshes lead to a reduction in the amplitude of $C_l$, decreasing the distance between the minimum and maximum points. In relation to $C_d$, presented in Figure 7c,d, it is evident that the mesh $512 \times 256$ overestimates the coefficient in relation to the other levels of refinement. There is a trend toward the convergence of $C_d \times t^*$ as the domain is refined, starting from the mesh $1024 \times 512$. For the simulated refinement levels, no convergence was observed when analyzing $C_l \times t^*$. Moreover, between $\alpha = 10°$ and $16°$, it is noted that the influence of refinement on the transient behavior of the coefficients is independent of the angle of attack.

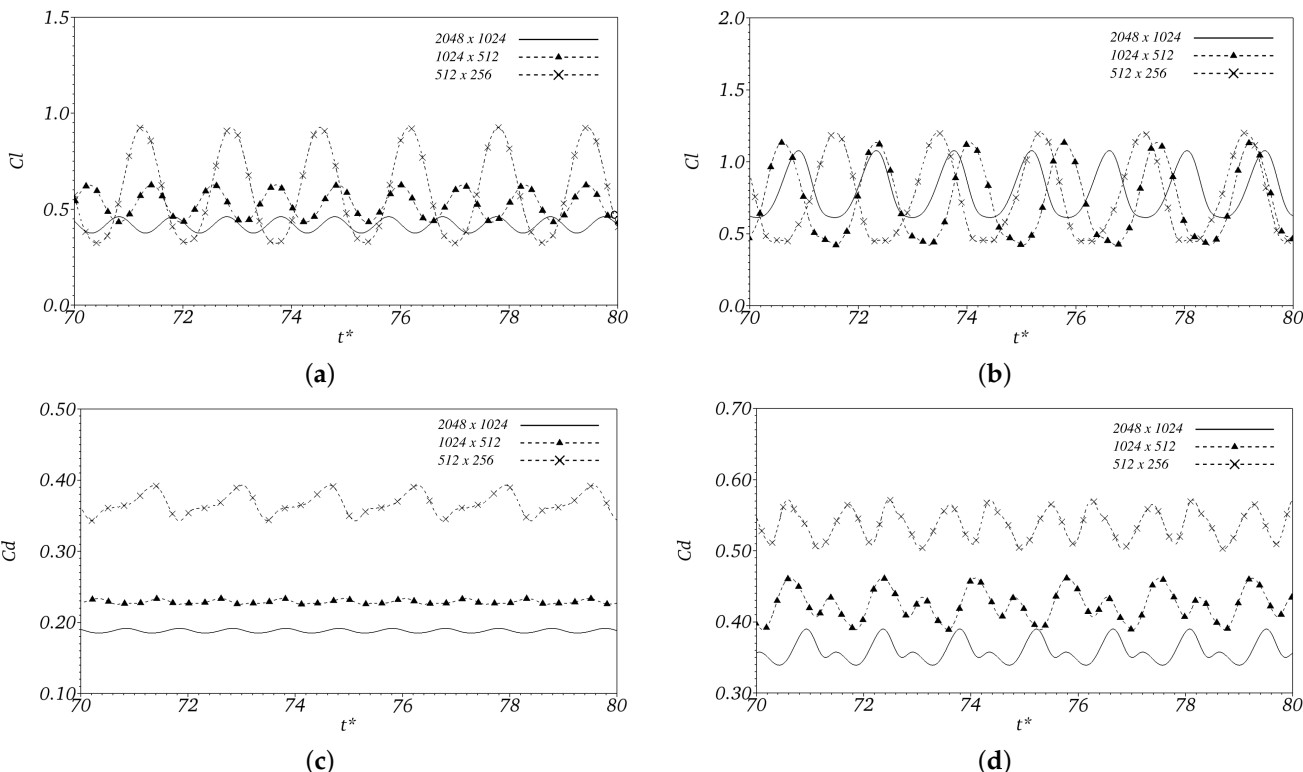

**Figure 7.** Influence of mesh refinement on $C_l \times t^*$: (**a**) $10°$ and (**b**) $16°$. Influence of mesh refinement on $C_d \times t^*$: (**c**) $10°$ and (**d**) $16°$.

The mean coefficients $C_l$ and $C_d$, to $\alpha = 10°$ and $16°$, are presented in Tables 1 and 2, respectively. It is observed that $C_l$ and $C_d$ tend to approach the reference results [42,43], from the mesh $2048 \times 1024$. The trend toward convergence observed in the transient behavior of $C_d$ is also observed in its average values as the domain is refined. In relation to mean $C_l$, to $\alpha = 16°$, there is an oscillation of the coefficient and an undefined behavior around the results of [42,43]. This indicates the need to perform one more simulation, increasing the level of refinement. This evidence is confirmed by the results presented by the mesh $4096 \times 2048$, which shows good agreement in relation to the reference works.

**Table 1.** Influence of mesh refinement: mean coefficients $C_l$ and $C_d$, obtained in the interval time $t^* = [70 : 100]$, for $\alpha = 10°$.

| Mesh | $C_l$ | $C_d$ |
|---|---|---|
| $512 \times 256$ | 0.5897 | 0.3670 |
| $1024 \times 512$ | 0.5264 | 0.2287 |
| $2048 \times 1024$ | 0.4173 | 0.1881 |
| [44] | 0.4184 | 0.1661 |

**Table 2.** Influence of mesh refinement: mean coefficients $C_l$ e $C_d$, obtained in the interval time $t^* = [70 : 100]$, for $\alpha = 16°$.

| Mesh | $C_l$ | $C_d$ |
|---|---|---|
| $512 \times 256$ | 0.7644 | 0.5369 |
| $1024 \times 512$ | 0.7209 | 0.4238 |
| $2048 \times 1024$ | 0.8023 | 0.3594 |
| $4096 \times 2048$ | 0.7100 | 0.3221 |
| [42] | 0.7263 | 0.3075 |
| [43] | 0.7594 | 0.3147 |

In Figure 8a, the order of spatial convergence is presented as a function of the relative percentage difference of mean $C_d$, calculated in relation to the work of [42]. To $\alpha = 10°$, a decay to the second order of convergence is observed. To $\alpha = 16°$, the order of convergence is diminished, tending toward the first order. This decrease can be explained by the intensification of the physical effects associated with the increase in the angle of attack, especially the detachment of the flow due to the strong adverse pressure gradient, which, in turn, promotes changes in the pressure field around the airfoil. The good representation of these phenomena depends on well-discretized domains.

The spatial convergence order can also be evaluated from the horizontal Lagrangian velocity. For this, the norm $L_2$ is calculated, used as a measure of numerical solution error,

$$L_2(U_i) = \sqrt{\frac{\sum\limits_{p=1}^{N_L} \left(U_i - U_i^{FI}\right)^2}{N_L}}, \tag{34}$$

where $U_i$ is the calculated Lagrangian point velocity and $U_i^{FI}$ is zero for flows over airfoils.

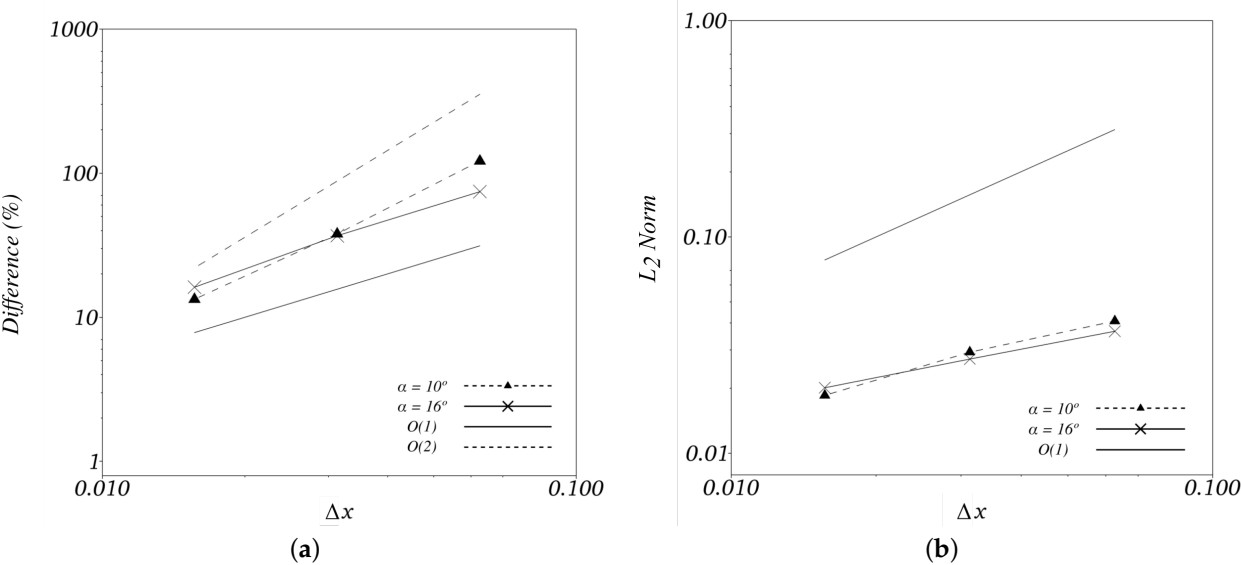

(a)  (b)

**Figure 8.** Order of spatial convergence. (**a**) Decay as a function of the relative percentage difference of mean $C_d$ in relation to the work of [42]. (**b**) Decay as a function of the time average of the $L_2$ norm of the horizontal Lagrangian velocity.

Note, as shown in Figure 8b, the decay of the norm $L_2$ tends to the first order of convergence $\alpha = 10°$ and $16°$. It is known that the Fourier pseudo-spectral method, when coupled with the immersed boundary method, loses accuracy, reducing the high order of convergence associated with spectral methods. Furthermore, the choice of the distribution and interpolation function is essential for the accuracy of the results obtained. There are functions that are more accurate than the "hat" function, such as the cubic function [45]. However, in the present work, the "hat" function becomes more appropriate due to the slenderness of the trailing edge of the airfoil.

From the results presented, it is observed that the need to propose simulations for one or two more levels of refinement greater than 4096 × 2048 is needed in order to guarantee not only the trend but the mesh independence in the proposed problem. However, the biggest limitation of these simulations refers to the high computational cost of modeling (CPU time). In simulations of flow over a step, for $Re = 400$, using the IMERSPEC methodology, [25] observes a linear increase of approximately five times in CPU time when the number of collocation points of the Eulerian mesh is increased by four times, which means that the computational time increases on the order of $NlogN$, as expected for the

Fourier pseudo-spectral method. Nevertheless, this includes the need for computers with excellent processing levels, including the need to use simulation parallelization techniques. Furthermore, by the $L_2$ norm (Figure 8b), a first-order spatial convergence decay is verified. This fact, associated with the high computational cost, is also responsible for reducing the cost-effectiveness of proposing simulations with very refined meshes.

### 4.3. Influence of the Number of NIT Interactions of Multi-Direct Forcing

To analyze the influence of the maximum number of interactions ($NIT$) of Multi-Direct Forcing, this paper proposes to analyze $C_l$ and $C_d$, varying $NIT = 1$ up to 100. The calculation domain has been refined to a mesh of $2048 \times 1024$, and the airfoil is positioned at $\alpha = 16°$.

The time evolution of $C_l$ and $C_d$ are shown in Figure 9a,b for different values of $NIT$. There is a notable difference in the transient behavior of $C_l$ and $C_d$ to $NIT = 1$ in relation to the others. Numerically, the analysis of the temporal evolution of $C_l$ and $C_d$ reflects the accuracy of transient modeling of fluid dynamic effects on the airfoil. Between $NIT = 30$ and $NIT = 50$, the convergence of the temporal evolution of $C_l$ and $C_d$ are obtained, including, in the initial moments of the flow, between $t^* = 0$ and $t^* = 6$. In this interval, in which the flow is not yet fully developed, a greater number of interactions is expected to guarantee the non-slip condition over the immersed boundary.

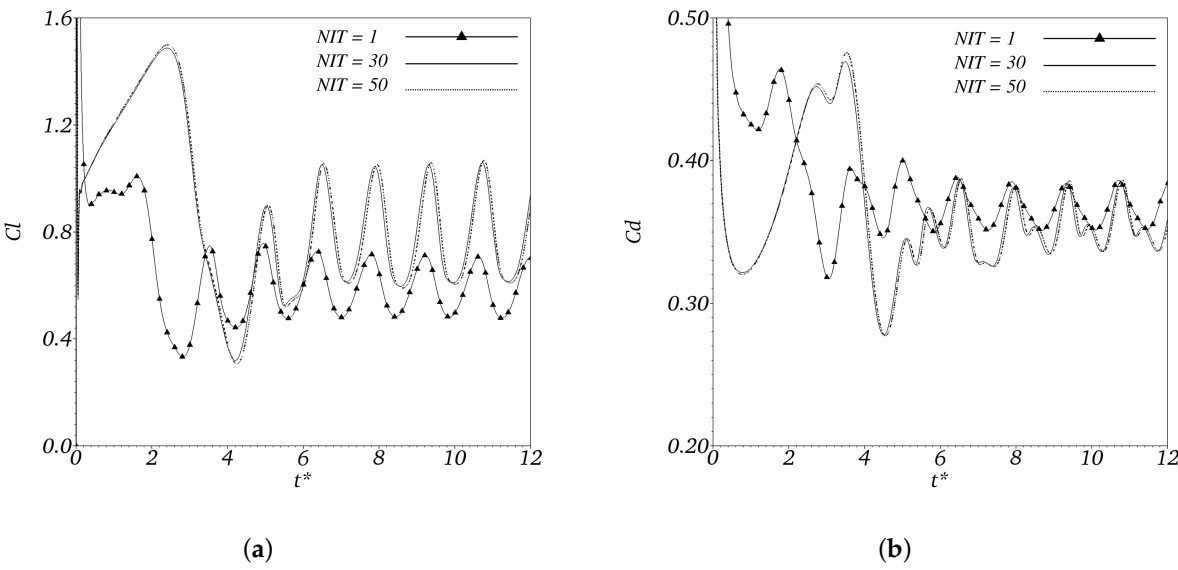

**(a)**　　　　　　　　　　　　　　　　**(b)**

**Figure 9.** Temporal evolution of (**a**) $C_l$ and (**b**) $C_d$ for different $NIT$ values.

The variation in means $C_l$ and $C_d$, obtained in the interval $t^* = [70 : 100]$, in the function of $NIT$, is shown in Figure 10a. The influence of $NIT$ on the results of $C_l$ is more noticeable than in relation to the results of $C_d$. Varying $NIT = 1$ to $NIT = 10$, it is evident that few interactions are already enough to reach the convergence of the average results of $C_l$, remaining practically constant from $NIT = 30$. The coefficient of $C_d$ undergoes a slight influence when varying $NIT = 1$ to $NIT = 10$. From then on, it remains constant for the entire simulated range. Taking the average into account, the transient effects of $C_d$ are attenuated for small values of $NIT$. It is observed that the divergence of the temporal evolution of $C_d$ to $NIT = 1$ in relation to the others, presented in Figure 9b, was not reflected in the average of $C_d$ at the same level of interaction. The convergence of average results does not guarantee the convergence of transient results.

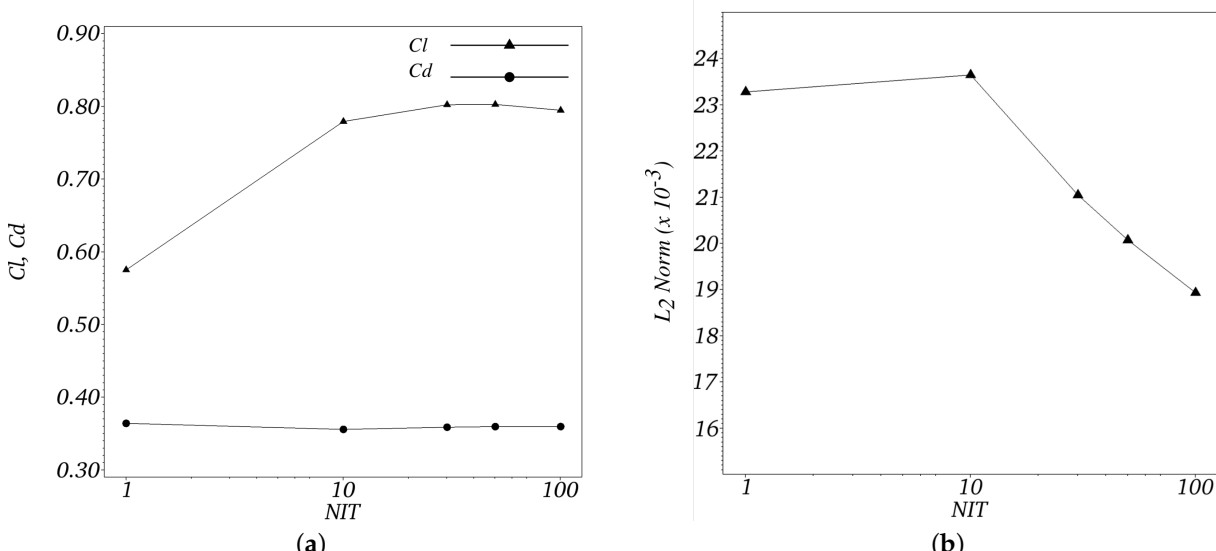

**Figure 10.** (**a**) Variation of mean $C_l$ and $C_d$ as a function of the number of $NIT$ interactions. (**b**) Temporal mean of the $L_2$ norm of horizontal Lagrangian velocity as a function of $NIT$.

The norm $L_2$ of the horizontal Lagrangian velocity is presented as a function of the variation of $NIT$ in Figure 10b. From $NIT = 10$, the norm $L_2$ decays to $NIT = 100$ almost linearly. The increase in $NIT$ reflects on the accuracy of imposing the no-slip condition on the immersed interface. Therefore, the Lagrangian velocity calculated at its respective Lagrangian point tends to get closer and closer to the velocity imposed at the airfoil boundary, reducing the norm $L_2$.

### 4.4. Influence of Angle of Attack

This section evaluates the influence of the angle of attack variation ($\alpha$) in the dimensionless coefficients of the airfoil and highlights the fluid dynamic phenomena of the flow by analyzing the fields of velocity, pressure and vorticity. The simulations are performed for the $2048 \times 1024$ mesh, and the maximum number of Multi-Direct Forcing interactions is $NIT = 50$. The mean of the coefficients is performed over the interval $t^* = [70 : 100]$, in which the flow is already fully developed.

The increase in $C_l$ as a function of $\alpha$ is shown in Figure 11a. There is good agreement with the results presented by [42,43]. In general, the increase in $C_l$ is due to the intensification of the low-pressure region in the upper forward part of the airfoil downstream of the leading edge, leading to increased lift force, $F_L$. From the curve presented, it is possible to determine the occurrence of the stall, a phenomenon where there is a sharp drop in the lift force and a drastic increase in the drag force. Note that the stall angle occurs for $\alpha = 28°$, in accordance with [43,46].

The present work proposes the simulation of flows with a low Reynolds number ($Re \leq 10^3$), and there is no direct imposition of disturbances and noise on the input boundary condition. These factors hinder the transition from the laminar boundary layer to the turbulent one over the surface, which leads, therefore, to the early detachment of the flow. Under these conditions, [12] shows, experimentally, that airfoils subjected to such flows do not stall. However, the immersed boundary method, when associated with the Fourier pseudo-spectral method, decreases the accuracy of the spectral method. The IMERSPEC methodology presents numerical and computational errors that intensify smoothly with the increase in $\alpha$, as shown in Figure 8. These errors are the source of noise in the solution, capable of leading to the flow transition and, consequently, to the appearance of the phenomenon, as observed in $\alpha = 28°$.

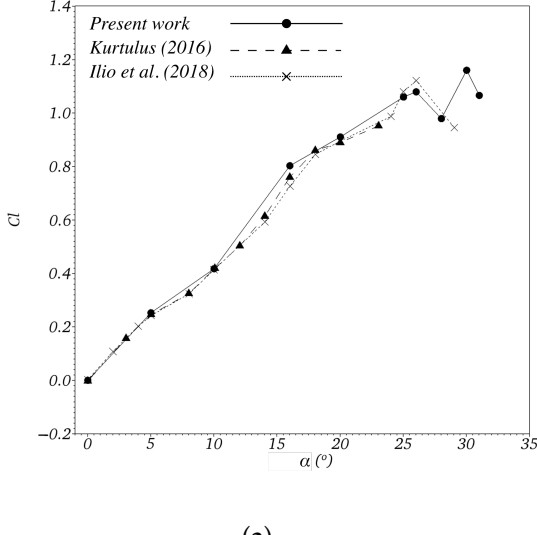

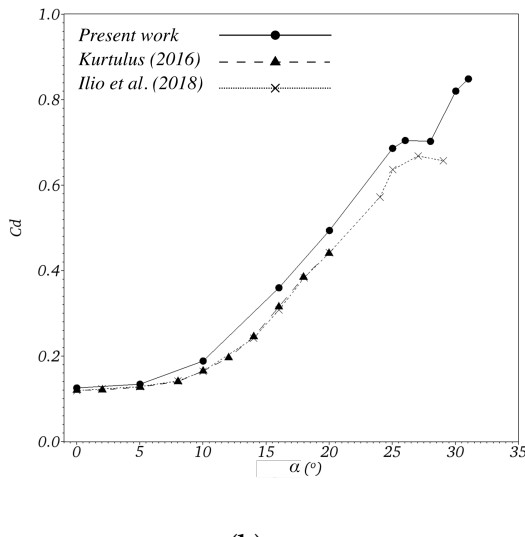

(**a**)                                                           (**b**)

**Figure 11.** (**a**) Mean $C_l$ variation as a function of attack angle ($\alpha$). (**b**) Mean $C_d$ variation as a function of the angle of attack ($\alpha$).

For $\alpha < 10°$, the drag force, $F_D$ is mainly composed of the frictional effects of the boundary layer, minimally influencing the increase in $C_d$, as shown in Figure 11b. As $\alpha$ increases, there is an intensification of the adverse pressure gradient downstream of the flow. Therefore, for $\alpha > 10°$, pressure drag due to surface flow detachment becomes dominant over the airfoil, leading to a significant increase in $C_d$. It can be seen that despite the good agreement, $\alpha > 10°$, a small distance is noticeable from $C_d$ regarding the results of [42,43]. The separation of the flow increases its complexity and demands greater accuracy from the methodology for the representation of the physical model.

In the pressure field for $\alpha = 0°$, represented in Figure 12a, it is observed that the pressure distribution is equal on the upper and lower surface, a condition that occurs due to the symmetry of the airfoil, subjected to a zero angle of attack. The low-pressure region in the extraction, just after the leading edge, evolves and intensifies with increasing $\alpha$, as shown in Figure 12b–d. In this region, the flow accelerates to pass and remains attached to the surface of the airfoil, reducing the pressure. Therefore, a net distribution of pressure is promoted, which is physically responsible for the appearance of $F_L$ in the upward direction. The increase in $\alpha$ shifts the flow separation point closer to the leading edge, causing the low-pressure region to occupy less space on the airfoil surface. For all angles, shown in Figure 12, a region of high pressure occurs at the stagnation point.

The streamlines of the mean velocity fields for different values of $\alpha$ are shown in Figure 13. For $\alpha = 0°$ (Figure 13a), the flow is aligned and does not show the formation of swirling structures. For $\alpha = 5°$ (Figure 13b), the flow shows the formation of the first recirculations on the suction side of the airfoil, close to the trailing edge. For this angle, therefore, part of the flow is already detached from the surface due to the adverse pressure gradient.

By the streamlines shown in Figure 13c–f, it can be seen that the increase in the angle of attack is responsible for the displacement of the flow separation point toward the leading edge. Therefore, the detachment of the flow and the winding of the separated shear layer is anticipated. This is one of the causes that justifies the linear increase in $C_l$, shown in Figure 11a and experimentally observed in [12]. The separation region tends to occupy a large part of the suction side, with an increase in size characteristic of the swirling structures that are formed and are, subsequently, advected upstream.

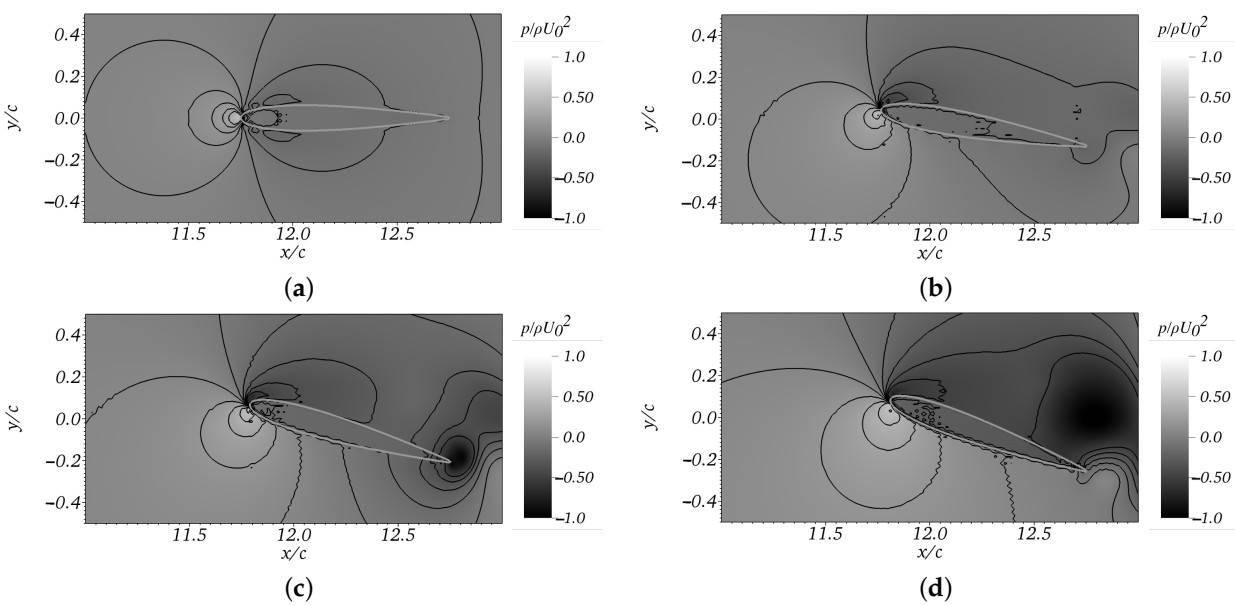

**Figure 12.** Isolines of pressure fields, at $t^* = 80$: (**a**) $\alpha = 0°$, (**b**) $\alpha = 10°$, (**c**) $\alpha = 16°$ and (**d**) $\alpha = 20°$.

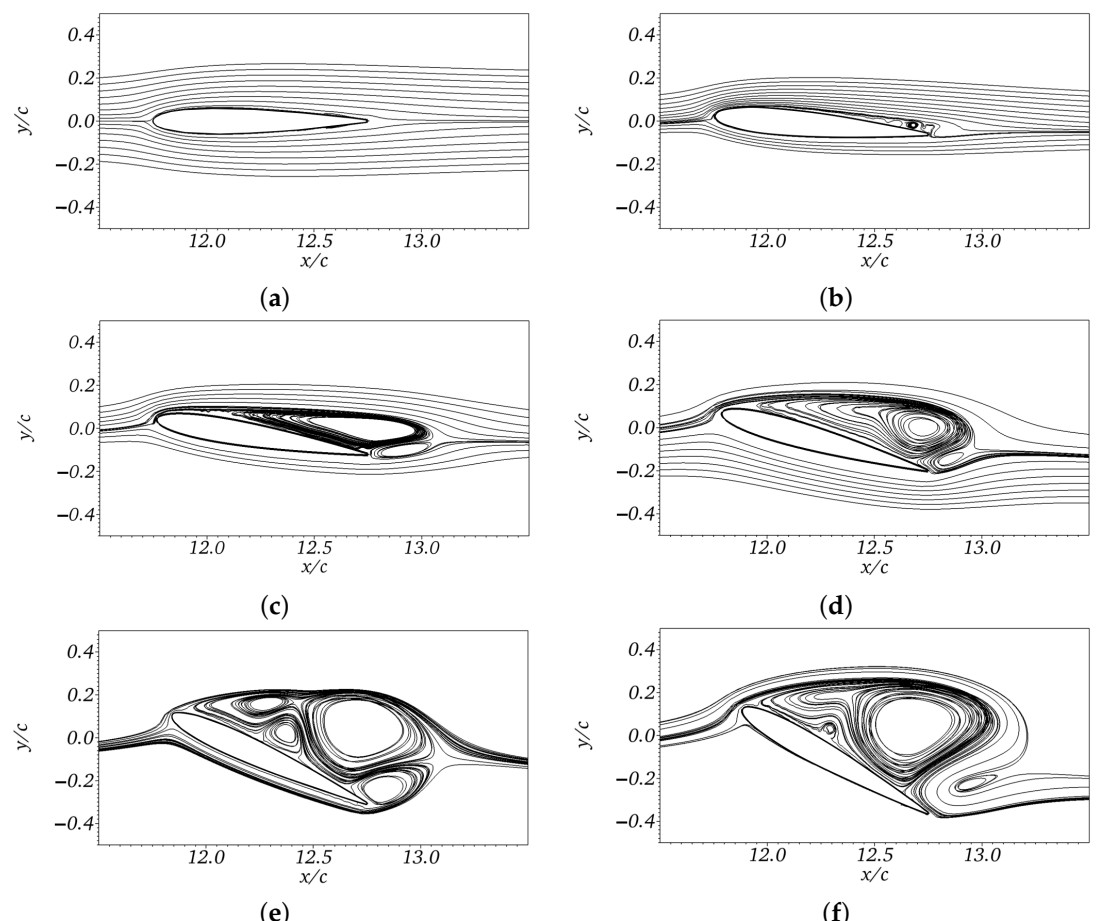

**Figure 13.** Streamlines of mean velocity fields: (**a**) $\alpha = 0°$, (**b**) $\alpha = 5°$, (**c**) $\alpha = 10°$, (**d**) $\alpha = 16°$, (**e**) $\alpha = 25°$ and (**f**) $\alpha = 30°$.

The frequency of the release of turbulent structures due to flow detachment is quantified by analyzing the Strouhal number, calculated by,

$$St = \frac{f_r c}{U_0} \tag{35}$$

where $f_r$ is the oscillation frequency of the maximum amplitude of $C_l$ in the spectral domain.

Figure 14 shows the variation in St for different angles of attack. The results presented are in good agreement with [43]. It is observed that $St$ is sensitized by the release of swirling structures, from $\alpha > 5°$, imposed by the intensification of the detachment of the flow. In $\alpha = 10°$, there is an overestimation of $St$ in relation to the reference work. Increasing the angle of attack leads to a decrease in $St$, and the frequency of release of the structures, responsible for the greater amplitude of $C_l$, is reduced. For $\alpha = 25°$, there is a decrease in $St$, followed by a smooth recovery to $\alpha = 30°$, due to the occurrence of the stall phenomenon.

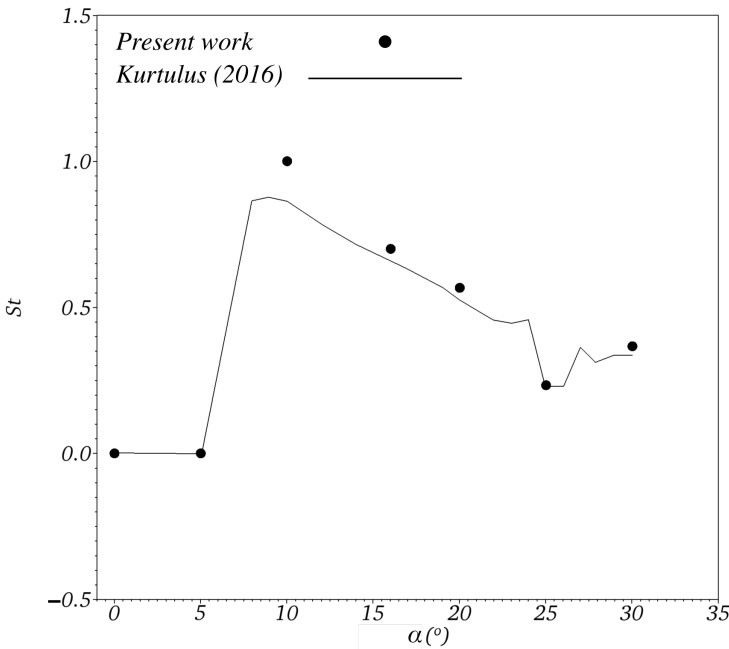

**Figure 14.** Variation of $St$ as a function of $\alpha$.

For $\alpha = 0°$ and $5°$, as shown in Figure 15a,b, respectively, the adverse pressure gradient is zero or minimal, keeping the flow attached to practically the entire surface of the airfoil. As a result, an aligned flow is obtained, with the absence of recirculations. For $\alpha = 10°$ (Figure 15c) and $\alpha = 16°$ (Figure 15d) it is possible to observe the detachment of a pair of swirling structures from the trailing edge, alternating, forming the von Kármán wake.

For $\alpha = 25°$ (Figure 15e), there is a wake pattern formed by two pairs of alternating structures, where each of the pairs, individually, forms independent wake regions with different characteristics downstream of the airfoil. It is a chaotic, transient wake pattern present in a short range of angles of attack. This justifies the sharp drop in $St$ to $\alpha = 25°$, as shown in Figure 14, evidencing the approach of the stall phenomenon. For $\alpha = 30°$ (Figure 15f), while pairs of counterclockwise swirling structures are detached downstream in a downward direction, a row of small clockwise structures is formed at the top.

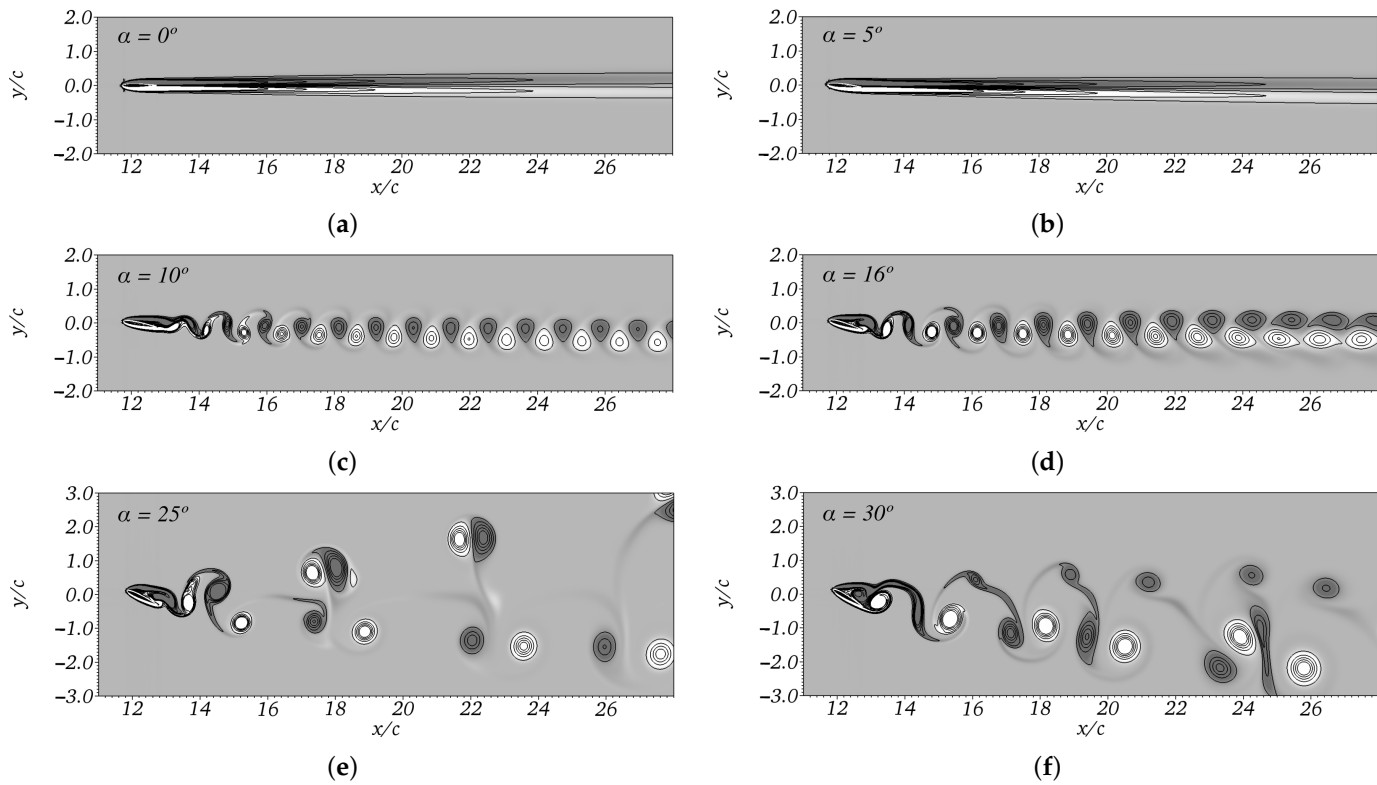

**Figure 15.** Instantaneous vorticity fields $-1 \leq w_z c / U_0 \leq 1$ over the NACA 0012 airfoil for different $\alpha$ values.

## 5. Flow over a Vertical Axis Turbine

The present work proposes the modeling of a two-dimensional and incompressible flow, using the IMERSPEC methodology, on a vertical axis wind turbine, represented by a three-bladed rotor. The blades are constituted by the NACA 0015 airfoil. The domain dimensions, dimensionalized by the airfoil chord $c = 1.00$ (m), are presented in Figure 16.

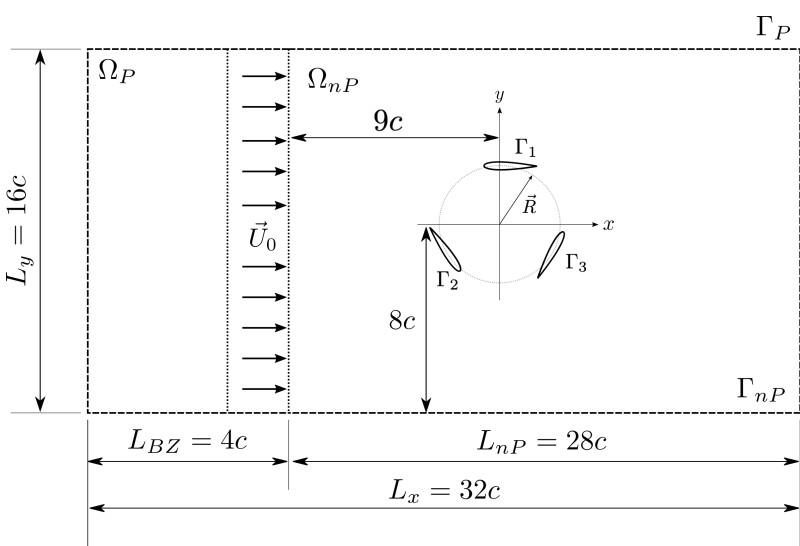

**Figure 16.** Calculation domain for the solution of flows over the vertical axis turbine.

It can be seen in Figure 16 that in the physical domain delimited by the boundary $\Gamma_{nP}$, there are three immersed boundaries, $\Gamma_1$, $\Gamma_2$ and $\Gamma_3$, formed by the Lagrangian points that model the airfoil NACA 0015. Each airfoil is discretized into $N_L = 300$ Lagrangian points. Together, these boundaries geometrically model the radius turbine rotor $R = 2c$. Under the

Lagrangian points, the no-slip boundary condition is imposed by the term $f_i(\vec{x}, t)$, from the calculation of the tangential velocity, given by Equations (25) and 26. The calculation domain (Eulerian domain) is discretized into $N_x = 2048 \times N_y = 1024$ colocation points.

The Reynolds number of the flow is equal to $Re = 100$, as proposed in the works of [20] and [33]. Therefore, the turbine is subjected to laminar flow, with free stream velocity $U_0 = 0.50$ (m/s), indirectly applied in the domain force zone. The kinematic viscosity of the fluid is calculated by $\nu = U_0 c / Re$, shown in (m$^2$/s). The specific mass of the fluid is equal to $\rho = 1.00$ (kg/m$^3$). In the interactive process of Multi-Direct Forcing, the maximum number of interactions was set to $NIT = 100$.

The turbine blades are subjected to a rotation speed of $\omega = 0.50$ (rad/s). Thus, the tip speed ratio (TSR) of the rotor is $\lambda = 2.0$. Regarding the time increment $\Delta t$, it is defined as $CFL = 0.1$. The simulations were performed on a computer with an Intel Xeon processor (E3-1270) with 3, 50 (GHz) of speed and 16.0 (GiB) of RAM memory.

In Figure 17, the coefficients of tangential force $C_t$ and normal force $C_n$ were analyzed, according to the azimuthal position $\theta$, calculated by Equation (31), on a single blade. The results were compared with work in [20,33]. By the coefficients obtained, the magnitude of $C_t$ turns out to be smaller than the magnitude of $C_n$. Furthermore, there is a greater sensitivity of $C_t$ about $C_n$ subject to more fluctuations. The oscillations and the small amplitude of $C_t$ about $C_n$, show the importance of applying numerical methods with a high order of convergence to solve problems of this nature. Through the IMERSPEC methodology, it is, therefore, possible to capture, with a good level of detail, fluctuations in $C_t$, suppressed by the results presented by [20,33] and hardly sensitized in real experiments.

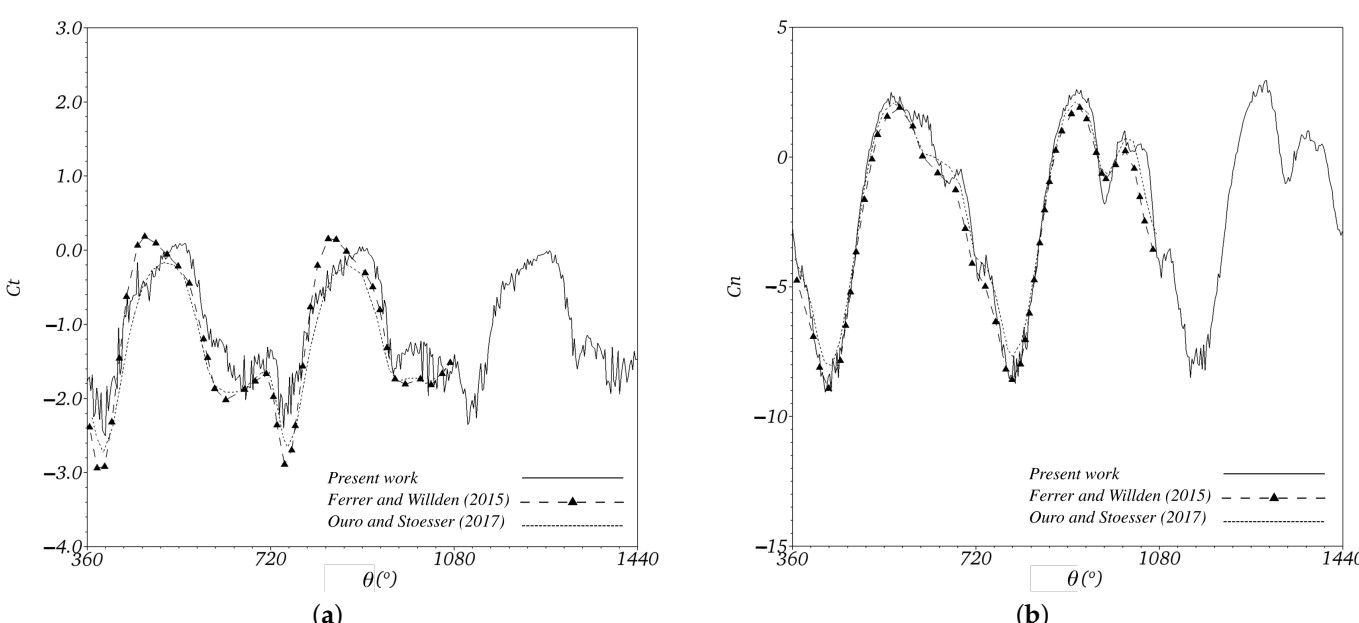

**Figure 17.** (**a**) Variation of $C_t$ as a function of azimuth position $\theta$ of a single blade. (**b**) Variation of $C_n$ as a function of the azimuthal position $\theta$ of a single blade.

It is observed in Figure 17a,b that the results of $C_t$ and $C_n$ show good agreement with the work of [20,33]. Even so, a slight divergence of $C_t$ in relation to the results of [20] is verified. In the downstream region ($540° \leq \theta \leq 720°$), there is a greater distance from the results of $C_t$ and $C_n$ in relation to the reference results. This behavior is due to the interaction effects between the blades and the wake region, which become intense in this portion of the rotor, which is also responsible for reducing the magnitude of $C_t$. Downstream, it is also observed that $C_t$ is subject to greater fluctuations. These variations are caused by a change in the pressure field around the blades, which is intensified by the release of turbulent structures arising from the detachment of the flow from the surfaces of the blades upstream, and which are advected to the center of the rotor. In the upstream region

$(360° \le \theta \le 540°)$, these effects are not intense on the blades, ensuring the convergence of the results in this portion.

The isolines of the absolute velocity field on the turbine rotor, at $\theta = 720°$, represented in Figure 18, are in good agreement with the velocity field presented by [33] under the same simulation conditions. A low-velocity region formed at the rear of the turbine rotor is observed due to the advection of swirling structures in the wake, originating from the detachment of the flow over the surface of the blades. Detachment becomes intense, especially for low Reynolds numbers, due to the low linear momentum of the flow. The blades are, therefore, subjected to a strong dynamic stall throughout the rotational motion. In addition, the low Reynolds number of the flow, proposed in this case study, prevents the promotion of positive mean values of $C_t$, impeding the factor of the ability of the simulated turbine to transform the kinetic energy of the flow into mechanical power.

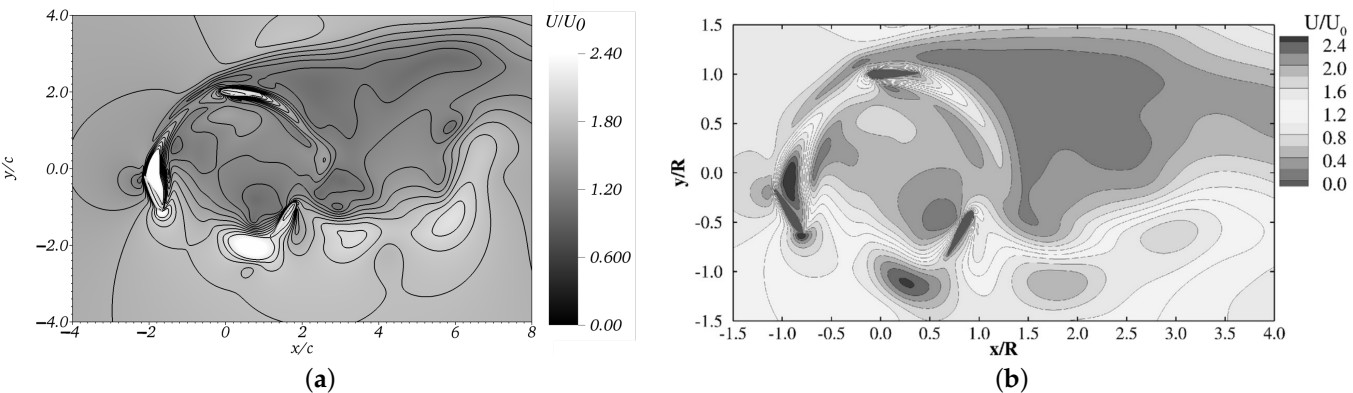

**Figure 18.** Isolines of the absolute velocity field over the turbine in $\theta = 720°$. (**a**) Present work. (**b**) [33].

## 6. Conclusions

In the present study, the application of the IMERSPEC methodology for numerical and computational simulation of two-dimensional flows over airfoils and vertical axis wind turbines was presented.

The validation procedure of the IMERSPEC methodology is based on the analysis of quantitative results and on the analysis of the physical phenomena derived from the modeling of flows over an airfoil NACA 0012 to $Re \le 10^3$. The dimensions analyzed, $C_l$, $C_d$ and the Strouhal number, show good agreement with the reference results from the mesh $N_x = 2048 \times N_y = 1024$. The average percentage difference in $C_d$ in relation to the reference works shows a decay to the second order of spatial convergence for angles of attack smaller than $10°$. It is observed, by the norm $L_2$ of the horizontal Lagrangian velocity, that decay to the first order of spatial convergence is independent of the angle of attack.

The analysis of the $NIT$ variation, used in the multiple force imposition procedure, reveals the importance of the interactive method since a few interactions are already enough to guarantee the convergence of the average and transient results of $C_d$ and $C_l$ in relation to the reference results. The increase in $NIT$ reflects a greater precision of the imposition of the no-slip boundary condition, observed by the decay of the norm $L_2$ from $NIT = 10$. The visualization of pressure fields, vorticity and streamlines for different angles of attack reliably models the expected physical phenomena.

The applicability of the methodology is extended to the modeling of two-dimensional flows over vertical axis turbine blades, now under the imposition of a rotational movement. To model the rotational movement of the turbine blades, a computational subroutine was developed based on a mathematical model, and added to the calculation platform of the IMERSPEC methodology. The procedure was carried out in two steps: updating the Lagrangian domain and calculating the tangential velocity imposed on the domain, as a function of the turbine rotational speed, at each time step.

For flows under low Reynolds numbers, $Re = 100$, the results of the tangential force and normal force coefficients, as well as the qualitative visualization of the flow through the velocity fields, are within the expected range in relation to other numerical methodologies. Therefore, it is indicated that the communication between the IMERSPEC methodology and the modeling of the rotational movement of the blades is being satisfied. The IMERSPEC methodology is, therefore, a potential and promising technique for solving problems of this nature.

More detailed investigations must be carried out to estimate the accuracy of the methodology for flows under high Reynolds numbers. The implementation of three-dimensional flows and turbulence models are options to be investigated, which are not available in the version proposed by the IMERSPEC methodology. However, it can be implemented for testing and obtaining future results.

**Author Contributions:** Conceptualization, L.M.M.; Software, F.P.M.; Validation, L.M.M.; Formal analysis, L.M.M.; Investigation, L.M.M.; Data curation, L.M.M.; Writing—original draft, L.M.M.; Writing—review and editing, F.P.M.; Supervision, F.P.M.; Project administration, F.P.M. All authors have read and agreed to the published version of the manuscript.

**Funding:** The research that led to these results was funded by FURNAS Centrais Elétricas and Research and Technological Development Program (P&D) of ANEEL.

**Institutional Review Board Statement:** Not applicable.

**Informed Consent Statement:** Not applicable.

**Data Availability Statement:** Data may be accessed by a request placed with either of the authors of this manuscript.

**Acknowledgments:** The authors would like to thank FURNAS Centrais Elétricas, the Research and Technological Development Program (P&D) of ANEEL and the Laboratory of Thermal and Fluid Engineering (LATEF) of UFG for supporting the development of this work.

**Conflicts of Interest:** The authors declare no conflict of interest.

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
