# Peer review of "Flow Modeling over Airfoils and Vertical Axis Wind Turbines Using Fourier Pseudo-Spectral Method and Coupled Immersed Boundary Method"

_axioms, doi:10.3390/axioms12020212_

Round 1
Reviewer 1 Report
The main objective of this work is to study flow over airfoils and vertical axis wind turbines. It is generally well written, but the authors should consider the following minor issues before the acceptance for publication.
1. Introduction part should be improved. The methodology for numerical modeling of 2D flows is well understood, and perhaps there are more references on this topic.
2. The work contains typographical errors, please correct. For example, description of Table 2.
3. When analyzing the influence of the mesh in Table 2, strange values ​​of the coefficient Cl are observed at 2048 × 1024. What are the reasons for such values?
4. When defining a parameter CFL, it is indicated that the values of this parameter are in the range from 0 to 1. Only the values 0.25 and 0.1 were used in the calculations? How this parameter is selected according to the type of flow?
Reviewer 2 Report
I believe the paper is of good scientific quality and deserves to be published in Axioms. I like the coupling between the Fourier pseudo-spectral method and the immersed boundary method. However, I would like to suggest some revisions before acceptance.
- I think the introduction can be improved by providing a more comprehensive description of the state of the art of numerical method for the problem of flow modeling over airfoils and wind turbines.
- I am not a native English speaker; however, figure labels should be translated into English (e.g., in Figure 8 Diferenca should be Difference, Norma should be Norm, and so on). Also, sen must be sin (for instance, in eq. 23).
- Line 137. I would prefer more details about the projection tensor to be able to derive the equations by myself (or references to papers published by other authors on the same topic).
- Section 2.3. could be improved. In particular, I would like to understand how the immersed boundary method allows enforcing the physical boundary conditions at the surface of the airfoil.
- I may be wrong; however, I do not see convergence in the results provided in Tables 1 and 2. Please comment on the convergence of the method.
